# Agile Comprehensive Care: A Pragmatic Trial of a Systemic Intervention for High Utilizers of Emergency Departments

**DOI:** 10.3390/healthcare13192391

**Published:** 2025-09-23

**Authors:** Melissa Casey, Dinali Perera, Hung Vo, Leilani De Silva, David M. Clarke

**Affiliations:** 1Department of Psychiatry, Monash University, Clayton, VIC 3168, Australia; dinali.perera@outlook.com (D.P.); hung.vo@monash.edu (H.V.); leilani.desilva@gmail.com (L.D.S.);; 2Agile Mental Health, Brighton, VIC 3186, Australia; 3Monash Health, Clayton, VIC 3168, Australia

**Keywords:** systemic intervention, psychological treatment, high utilizers, emergency department presentations

## Abstract

**Background**: The agile Comprehensive Care (aCC) is a service that was developed in response to a detailed analysis of the most frequent attendees to Monash Health’s emergency departments (ED). Analyses revealed a group of clients with complex mental health issues who were receiving disintegrated care resulting in suboptimal clinical outcomes, high demand on resources and substantial costs. In a real-life setting, we sought to evaluate, through follow-up, the effectiveness of the aCC service, which aimed at stabilizing the system’s response to high utilizers by developing comprehensive service plans, modelled on the General Psychiatric Management Framework, to be utilized by all clinicians no matter where the person presents within the system of care. **Methods:** A single group pre/post study was undertaken involving the follow-up analysis of 27 patients discharged from the aCC clinic after intervention. A comparison of 12-month median pre-aCC and median post-aCC service utilization and service costs was undertaken using Wilcoxon signed rank tests with effect sizes reported as r. In addition, we received feedback from staff within the health service who received support from aCC for their complex clients. **Results:** ED presentations decreased significantly from the pre-intervention period to the post-intervention by a median of 15 visits to a median of four visits. Mental health ED presentations decreased from a median of nine visits to one visit. Median service costs decreased from AUS 64,921 to AUS 19,329. aCC support gave staff greater confidence in working with this complex group. **Conclusions:** agile Comprehensive Care, involving the development of a systems-wide treatment plan coupled with patient and clinician support, improved outcomes and reduced service usage and costs for a complex group of high utilizer patients.

## 1. Introduction

High utilizers of emergency departments are often defined as those having four or more visits per year [1]. Sometimes referred to as high-flyers and super-utilizers, this group represents a heterogeneous group with complex health needs. While high utilizers consist of a small percentage (4.5–8%) of ED attendees, they account for a disproportionately large number (21–28%) of ED visits and associated costs [1,2,3]. High utilizers of the ED create a significant burden for ED services. They are associated with increased cost and many adverse outcomes, including increased mortality [3,4,5,6]. Those who present and receive unnecessary services may also be vulnerable to iatrogenic injury or unintentional harm [7], including errors related to poor transitional care and multiple handoffs [4,8]. A high prevalence of mental illness is a striking commonality seen amongst high utilizers globally, across diverse healthcare systems [9]. Research indicates that 3.5–9% of presentations to public hospital EDs are from people seeking mental health (MH) care [10,11]. The trend of increasing numbers of MH presentations to the ED over time is visible in Australia [4], as elsewhere, with the literature indicating a growth rate of MH presentations twice that of overall presentations [12].

Psychiatric diagnoses most seen in high utilizers include depression, schizophrenia, somatoform, and personality and substance-use disorders [4,10,13,14,15,16,17,18]. These frequent attendees with a mental health diagnosis often present with suicidal ideation, self-harm, feeling unsafe, situational crises, depression, anxiety, and substance abuse [19], as well as an absent psychotherapist or a need for medication [20]. High utilizers view their visits as both necessary and unavoidable [5], while ED clinicians often view their presentations as time-consuming and unnecessary. This results in a tendency for high utilizers with mental health difficulties to be stigmatized [5,21]. Patients may feel unsatisfied when they are discharged without being admitted or offered treatment [22], and interpret their discharge as poor treatment [5]. As such, there is a pressing need for comprehensive interventions that target the complex medical, psychological and social needs of these ED attendees.

Case management/assertive community outreach programs are commonly employed for high utilizers and consist of an interdisciplinary approach that coordinates available care services, whilst providing continuous monitoring and evaluation of progress with the intention of promoting both patient health and system outcomes [1]. Similarly to case management, individualized care plans involve a multidisciplinary approach to tailor care for the complex medical and social needs of patients and have been found to reduce hospital admissions and re-admissions, and hospital costs for complex high-utilizing patients [23]. There is, however, limited research exploring the effectiveness of these interventions for improving ED use by those with mental health conditions [24]. A review conducted by Dieterich and colleagues [25] found that intensive case management, compared to standard care, may reduce hospitalization and improve patient outcomes for those with serious mental illness and a high level of service use; however, the evidence supporting this was deemed low-to-moderate quality. It has been documented that high ED utilizers with mental health issues are most in need of long-term therapeutic support [26] and can benefit from psychological intervention [4,27,28]. Baker and colleagues [27] found that for high utilizers, a case management approach consisting of individualized care plans and psychological intervention is effective in reducing hospital costs.

The research evidence for psychological treatment of high utilizers comes from personality disorder research [29]. Clinical guidelines and systematic reviews based on over 25 randomized controlled trials (RCTs) have shown strong support for structured psychological treatments for patients with BPD [30,31,32,33], including cognitive-behavioral and dynamic interpersonal therapies [29,34]. Due to few differences in overall treatment efficacy amongst the different psychological treatments evaluated [34], there has been an increasing focus on core features (called common factors) of therapy shared by the approaches evaluated [35]. Specifically, common strategies seen as central to the efficacy of BPD treatment include clear treatment framework, attention to affect, focus on treatment relationship, an active therapist, and exploratory and change-oriented interventions [36]. There is generally limited research exploring the effectiveness of psychological interventions for improved ED use by those high utilizers with mental health conditions.

The agile Comprehensive Care (aCC) is a service developed in response to a detailed analysis of Monash Health’s highest utilizers of emergency departments both at a quantitative and individual patient experience level. Analyses revealed a disintegrated service system for ED high utilizers. Specifically examining each patient’s pattern of presenting over time, it became evident that high utilizer patients received inconsistent service system responses, resulting in suboptimal clinical outcomes, an increasing trend of presenting, and increased pressure within the emergency department workforce due to the increasing demand on resources and substantial costs [4]. A high prevalence of BPD and other psychiatric diagnoses was also noted in this group. Despite a high degree of contact with services, this particular cohort of high utilizers made minimal gains, experienced iatrogenic harm from services, and their needs remained unmet [4]. The agile comprehensive care (aCC) service was developed with the objective to first stabilize the service system by developing a comprehensive care plan for the patient and providing capacity building, coaching and support for the clinicians at the frontline to enact the plan, and second, to encourage and support those frontline clinicians to then channel the high utilizer into aCC where clinicians employ empirically validated psychological therapies based on the common factors approach. Elements of intensive case management, the use of individualized care plans and systemic management of patient care pathways are part of the comprehensive therapeutic response utilized by aCC.

The present study aimed to evaluate a treatment program tailored for high utilizers of emergency departments within an urban hospital system in Melbourne, Australia. Monash Health has three main emergency departments. It was hypothesized that the provision of a comprehensive and coordinated therapeutic response to high utilizers would result in decreased ED presentations, service usage, iatrogenic effects, and service costs and result in patients leading more fulfilling lives. We anticipated this service would also be experienced as beneficial by staff who received support for their complex patients.

## 2. Materials and Methods

### 2.1. The Model of Agile Comprehensive Care (aCC)

The aCC team comprises four senior psychologists, a consultant psychiatrist, a research and administrative staff member and a My Care Pathway Coordinator. The My Care Pathway Coordinator is a dedicated senior nurse who manages the patient’s care pathway across the multiple sectors involved. The aCC team is based in Dandenong Hospital, Monash Health and provides services to patients and teams across all sites at Monash Health. aCC services are offered to mental health patients above 18 years of age. Services are offered to those patients who are amongst the top 20 most frequent presenters to Monash Health ED. This is identified through extraction of the top 20 most frequent presenters to ED by Monash Business Intelligence every 4 months. An in-depth file review of medical records is conducted by the aCC team on each patient to formulate the problem and determine if they would be suitable for an aCC intervention. The inclusion criteria were as follows: (a) patients with a history of frequent mental health presentations to crisis services over the past 5 years; (b) the patient’s presentation is severe and complex, suggesting systemic discord; and (c) aCC services will be able to add value to the patient’s existing treatment. In some cases, treating teams of patients who are complex and have problematic and chaotic presentations have also approached aCC for assistance with a management plan or general input and consultation. Each patient is discussed amongst the team to determine suitability for aCC and whether they would benefit from aCC intervention.

Frequent utilizers of emergency services often have multiple services involved in their care (acute, in-patient and community treatment teams, as well as police, ambulance and non-governmental organizations (NGOs)). aCC works systemically with these providers when needed, facilitating more cohesive patient care and better clinical pathways for patients through the development and enactment of a comprehensive service plan (CSP). The CSP is designed to provide an understanding of the high utilizer and develop a therapeutic response for the individual through clarifying the roles and actions required of multiple treatment providers under various circumstances. The template for the CSP is included in Appendix A. A CSP is predominantly developed to provide consistency and coordinated treatments for high utilizers with complex needs. Support of the teams and staff through secondary consultation and team supervision is also provided.

Patients are also offered direct individual psychological treatment where this is deemed suitable and if the patients are willing and able to engage in therapy. Therapy offered is dependent on patient needs, treatment history, patient preferences, and team capacity. This typically involves long-term psychotherapy over 12–24 months, as well as psychiatric review. aCC employs empirically validated psychological therapies, including cognitive analytical therapy, mentalization-based therapy and psycho-dynamic therapy–with a focus on the core features that all evidence-based therapies share [36]. Prior to the commencement of individual treatment, verbal and written informed consent is obtained. aCC clinicians delivering psychological treatment to patients also work systemically with their relevant providers to facilitate more cohesive patient care overall.

A Clinical Review meeting is conducted weekly under the guidance of a Consultant Psychiatrist for the purposes of reviewing patients’ progress.

### 2.2. Study Design

A single group pre/post study design was adopted to compare relevant service usage and costs in the 12 months prior to patients registering into the aCC clinic and 12 months following the patient discharge date. The study sample included all patients who were referred to and discharged from the aCC clinic after receiving either (a) systemic intervention alone or (b) systemic intervention plus specialist individual psychological therapy, and for whom there was a full 12 months of follow-up data.

### 2.3. Data Collection

aCC regularly measures the patient’s service usage, the patient’s experience and clinical outcomes. We excluded reporting on clinical outcomes in this paper, given the large amount of missing data points. Obtaining pre/post patient-reported clinical outcomes data alongside delivering the clinical care is not a part of standard clinical care, so we embedded processes that provided oversight to compliance with the clinical outcomes data collection; this governance involved monthly meetings reviewing compliance and following up. Whilst significant progress was made over time, the amount of missing data points necessitated exclusion from the analysis reported here.

Demographic, registration-related and clinical data were recorded about patients registered at the aCC clinic. Demographic data recorded contained date of birth, age at registration and gender. Registration details were captured, such as the referral source, registration date, discharge date and the type of intervention the patient received. Clinical data included the primary diagnostic category, other diagnostic comorbidities and the allocated clinician. aCC patient data were recorded in a standardized template in Microsoft Excel.

ED presentation and costing data were extracted from Monash Health’s data warehouse, which is a repository that integrates data from multiple business and health information systems. ED presentations were extracted for all patients that were registered with the aCC. Multiple variables were retrieved, such as, though not limited to, demographic variables (e.g., patient age at admission, gender), presentation-related variables (e.g., arrival date, departure date, triage category, presenting problem), diagnostic variables (e.g., primary diagnosis, primary procedure undertaken) and information about adverse events and incidents. Costing data provided costs across a range of service types, such as emergency presentations, hospital admissions, pathology and community services.

### 2.4. aCC Internal Program Feedback

Feedback was sought from other staff in the health service who worked alongside aCC. This was circulated via a Survey Monkey link/email by aCC clinicians. The primary purpose of this feedback was to inform future processes following an annual team planning and service improvement day. The survey was voluntarily and anonymous. It consisted of 10 questions and took approximately 5 min to complete. Examples of questions included the following: “What was the most helpful thing aCC did?”, “What was the least helpful thing aCC did?”, “Has contact with aCC changed your practice?”, “How useful is the CSP in guiding your treatment of the client?”

### 2.5. Study Sample

Twenty-seven patients met the inclusion criteria described above, with 12 months of follow-up data available. Most were female (*n* = 21, 77.78%), with a mean age of 36 years (standard deviation (SD) = 11.73). All patients received the systemic intervention; and specifically, 12 had CSPs developed and implemented. Six patients and their treating teams received support from the My Care Pathway Coordinator. Seven of the 27 additionally received individual psychotherapy. Of this group, four had a primary diagnosis of borderline personality disorder (BPD), one major depressive disorder, one dependent personality disorder, and one an eating disorder. Two of the BPD patients also had comorbidities including PTSD, schizophrenia, and substance-induced psychosis. The mean duration of treatment was 8 months.

### 2.6. Data Analysis

Non-normal distributions were detected in most of the measures included in this analysis. Due to this, Wilcoxon signed rank tests were undertaken to determine whether differences between pre-intervention and post-intervention periods were statistically significant. These tests were conducted on both service utilization and service costs recorded in the 12 months preceding the patient’s registration date into the aCC clinic and 12 months following the patient’s discharge date from aCC. Effect sizes (r) were calculated for Wilcoxon signed rank tests by using the formula suggested by Tomczak and Tomczak [37]. Conventions for effect sizes consider 0.5 as a large effect, 0.3 as a medium effect and 0.1 as a small effect [38]. An alpha level was set at *p* < 0.05 for statistical significance. All data processing and statistical analyses were conducted using the R version 4.0.3 [39] for Windows. Statistical packages included the use of rstatix 0.7.0 [40].

This study met criteria for being an operational quality improvement activity and was approved as such by Monash Health’s Research Support Services (Reference Number: RES-18-0000-008Q) and deemed exempt from formal review by the Monash Health Human Research Ethics Committee.

## 3. Results

### 3.1. Pre/Post-Intervention Service Utilization for Discharged Patients

Total ED presentations recorded in the pre-intervention period were 436, and in the post-intervention period were 338, which constituted a reduction of 22.5%. Total mental health ED presentations decreased from 297 to 189, representing a percentage change of 36.4%. All other mental health service measures demonstrated a pre-to-post reduction in total volume. The results do not appear to be influenced by seasonality. Inspection of emergency department presentations by month, across summer and winter periods, did not highlight any discernible seasonal patterns.

Patients presented at the emergency department a median of 15 times (IQR 8.5–20.5 times) in the 12-month pre-intervention period prior to their aCC treatment (Table 1). This decreased to a median of four times (IQR 0–15.5 times) in the 12-month post-intervention period. A Wilcoxon signed rank test demonstrated that this difference in ED presentations between the pre-intervention and post-intervention periods was statistically significant, *p* < 0.05. The effect size (r = 0.48) calculated was a medium effect. Similarly, mental health ED presentations decreased significantly from the pre-intervention (Mdn = 9, IQR 5.5–16.5) to post-intervention (Mdn = 1, IQR 0–7.5) period, *p* < 0.05. The effect size also was considered medium, r = 0.46. Reductions in the remaining service utilization measures were all statistically significant (Figure 1).

### 3.2. Pre/Post-Intervention Service Costs for Discharged Patients

Total service costs were AUS 2,707,069 in the pre-intervention period and AUS 859,428 in the post-intervention period, representing a 68.3% reduction for all patients included in this sample. Total emergency medical costs reported in the pre-intervention period were AUS 226,311, and in the post-intervention period, AUS 157,276, which constituted a reduction of −30.5%. Total emergency nursing costs decreased from AUS 219,145 to AUS 157,967, representing a percentage change of 27.9%. All other service cost measures demonstrate a pre-to-post reduction in costs.

Table 2 summarizes the changes in pre/post median service costs after aCC treatment. Total costs associated with pre-intervention services (Mdn = AUS 64,921, IQR AUS 30,174–AUS 141,967) used by patients decreased substantially when compared to total costs associated with post-intervention services (Mdn = AUS 19,329, IQR AUS 3335–AUS 30,713), *p* < 0.0001. Substantial reductions were observed for emergency medical (Mdn = AUS 7010, IQR AUS 4355–AUS 9929 to Mdn = AUS 1998, IQR AUS 0–AUS 6044) and emergency nursing costs (Mdn = AUS 6773, IQR AUS 4690–AUS 10,097 to Mdn = AUS 2247, IQR AUS 0–AUS 6085) between the pre- and post-intervention periods. These reductions were both statistically significant, *p* < 0.01. The effect sizes for these types of costs were medium, respectively, r = 0.55 and r = 0.55. Notably, other mental health costs significantly (*p* < 0.001) declined from a median of AUS 12,681 (IQR AUS 4882–AUS 25,809) to AUS 3021 (IQR AUS 427–AUS 9136). All other costs demonstrated a statistically significant reduction except for community costs (Figure 2).

### 3.3. aCC Internal Program Feedback from Within the Health Service

Twenty-five responses were received in total, and staff were from a number of services across the organization, including from Continuing Care Teams (CCT, *n* = 8), Inpatients Unit (*n* = 5), Emergency Psychiatric Services (*n* = 6), Mobile Support Team (MST; *n* = 2), Young Persons Mental Health Service (*n* = 2), Youth Consultation and Treatment Team (YCTT), Mental Health Forensic Assessment and Consultation Team (MH FACT), and Police Ambulance and Clinical Early Response (PACER) (all *n* = 1). Seventy-two percent of respondents reported having had between one and six patients who received support from aCC, 4% had numerous patients (*n* = 1), and 8% completed the survey without having any clients with aCC (*n* = 2).

The majority reported their contact with aCC as consisting of support with the development of a CSP (Comprehensive Service Plan; *n* = 14, 60.87%), shared care of the patient (12 people = 52.17%) or a secondary consult (12 people = 52.17%). Many respondents reported having found aCC’s CSP development work as the most helpful aspect of the service’s involvement, with the majority reporting that the CSP was useful (73.91% extremely useful, 17.39% somewhat useful) in guiding their treatment of the patient. Additionally, 95.24% reported following and enacting the CSP. It was reported that this work helped to “support” and provide “direction” to the treating services in “managing [these] clients in crises”, “supported a consistent approach” within teams, and aided in “reduce [ing] patient’s reliance on ED or inpatient admissions to get their needs met”. aCC’s input around “formulation”, client “risk”, as well as “what the service can do to support them” [the patient] were reported as helpful. According to one staff member, aCC support “provide [d] the client with high quality care and interventions that have minimized clients’ risks, improve [d] client functioning and mental state…”.

Most clinicians (65.22%) reported that contact with aCC had changed their practice in some way. Overwhelmingly, clinicians reported a more “consistent” management of clients that took place via a lens of “understanding” and “support” for the client’s safety. Clinicians felt they were now able to work more cooperatively and with “resilience”, and to think about their clients “in more complex ways”. Between-team coordination and consistency in treatment delivery were also reported. The accessibility of specialist psychological treatment for this group was also reported to be beneficial. It was reported that aCC support has allowed clinicians to “address the client’s needs in more therapeutic ways” and feel more confident with holding some positive risk.

## 4. Discussion

Baker and colleagues [27] found that individualized care plans and psychological interventions reduce hospital costs. The current study adds further insight into the additive value of also stabilizing the system’s response through a comprehensive care plan (the ‘know-what’), supporting the many frontline staff who clinically respond to high utilizers (‘know-how’) as well as providing longer-term psychological treatment. Indeed, our experience is that stabilizing the systems response and the resulting patient experience of that consistency over time provides the foundation for the patient being able to move past receiving the crisis response (only) and move into longer-term therapeutic treatment.

Central to the aCC model working was the my care pathway role. This role provided the relational and linked-up care from multiple services, often drawn upon by the high utilizer (acute, inpatient, community treatment teams, police, ambulance, prisons, NGOs).

After the patients had received an aCC episode of care and been discharged from the service, total mental health ED presentations declined by 36.4% in real terms. Patients in our sample presented a median 15 times in the 12-month period prior to their aCC episode of care and then only four times in the 12-month period after their discharge, so the effect of their aCC intervention held value. Costs similarly declined. Total service costs were AUS 2,707,069 in the pre-intervention period and AUS 859,428 in the post-intervention period, representing a −68.3% reduction.

Frontline clinicians working with high utilizers anecdotally report demoralization, frustration and lack of skill in knowing how best to treat them giving their re-presenting behavior, so this study also focused on the effect of building up and supporting frontline clinicians in their treatment of high utilizers. Of staff surveyed, 73.91% found the aCC development of the comprehensive service plan extremely useful, as it provided practical guidance on how to best treat the high utilizer in crisis. Additionally, 65.22% found aCC had influenced their clinical practice to be more understanding and consistent whilst at the same time allowing them to think about their patient functioning in more complex ways; they were better able to hold clinically indicated risks, and having aCC staff involvement reduced their personal anxiety in the treatment of high utilizers.

### Limitations

The findings of this study should be interpreted with consideration of its limitations. The low participant numbers suggest caution is required in generalizing conclusions. Nevertheless, the study had similar numbers compared to previous evaluations of similar programs, and the general findings were consistent with those of previous studies [41,42,43]. The aCC clinic is still operational, and this study reports on its impact relating to patient outcomes, service utilization and cost. As more data are collected, there is potential to validate the effects on a larger sample size in the future. Practice-based research in public healthcare contexts faces many methodological issues. In any public health setting, the formation of a control group is almost impossible or, in most cases, unethically practical. The absence of a control group enforces stress on the internal validity of the presented study, threatening valid conclusions from the reported results [44]. Patients may reduce their presentations to the emergency department over time and could potentially reflect regression to the mean and natural regression regardless of intervention [45,46]. The lack of a control group is a limitation in this study, as we could not determine the impact of potentially natural attrition on the aCC intervention effect. However, one recent Australian 10-year longitudinal study [47] has indicated that patients with psychiatric illnesses were more likely to be ongoing (persistent) frequent presenters. This current study focused on patients with complex psychiatric conditions, but future investigations should incorporate control groups to incorporate persistence of presentations and illuminate intervention effect(s). Another consideration is that only data obtained from the hospital data warehouse were included. It is likely, given the complexity of the cohort, that some clients may have occasionally sought healthcare from other hospitals. These data were not obtained due to resource limitations. Furthermore, this study focused on a metropolitan service catchment, and therefore, intervention effects reported may not be generalizable across rural regions. This study only examined service utilization and service costs 12 months preceding and following the intervention. It is not clear whether these outcomes will be sustained over a longer period, and risk of relapse may be a concern with these patients. Previous studies have utilized follow-up periods ranging from 6 to 12 months post-intervention [16]. Furthermore, the reason for emergency department presentations was not collected. This limited our ability to extend the analysis to explore changes in service utilization costs and prevented differentiation between high- and low-acuity presentations, which can vary in resource use and cost. Future research should assess long-term outcomes of intervention effectiveness. Given the large amount of missing data, this study was unable to report on clinical outcome measures data. This would have been useful to additionally measure the patient experience and improved clinical outcomes. This is particularly important and valuable information, and future studies are encouraged to capture these data where possible.

## 5. Conclusions

An integrated and comprehensive service approach offering individual and systemic intervention for those who frequently present to emergency departments is an effective approach to improving clinical outcomes for this group, reducing their overall reliance on services and reducing hospital services and costs that are not adding value.

## Figures and Tables

**Figure 1 healthcare-13-02391-f001:**
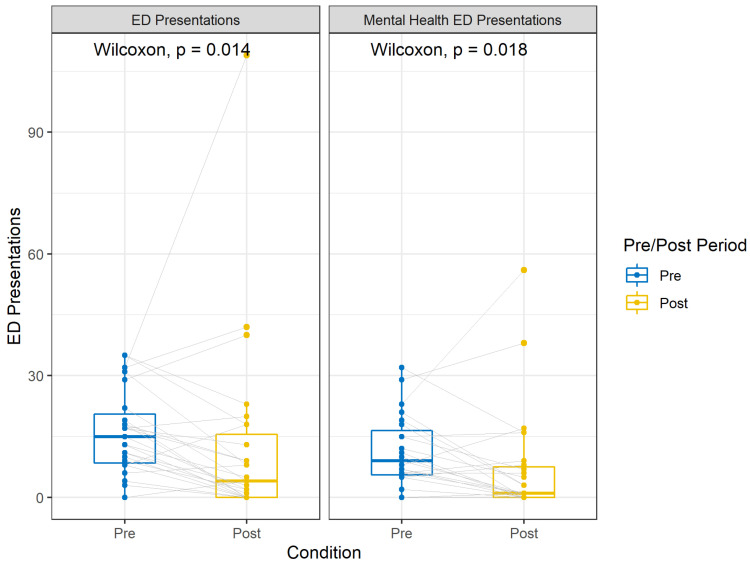
Boxplots of 12-month pre/post-intervention median ED presentations by patients discharged from the aCC program.

**Figure 2 healthcare-13-02391-f002:**
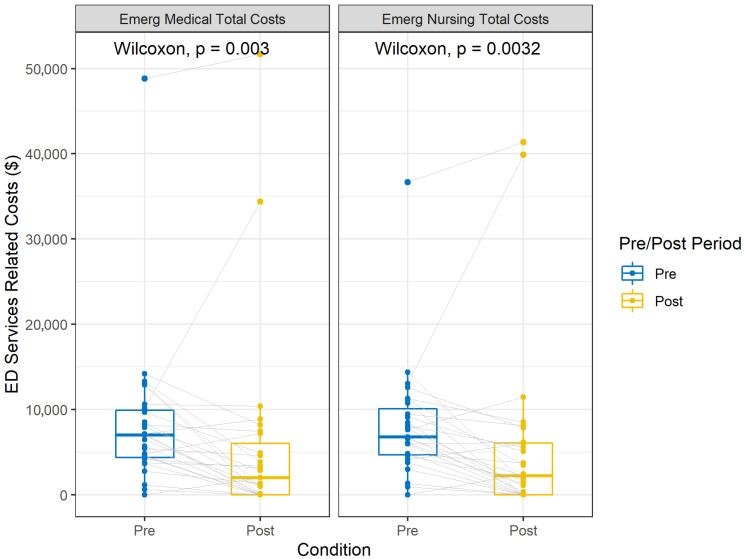
Boxplots of 12-month pre/post-intervention median ED services-related costs (AUS) for patients discharged from the aCC program.

**Table 1 healthcare-13-02391-t001:** Twelve-month pre/post-intervention median service utilization by patients discharged from the aCC program.

Measure	*N*	Pre-aCC (IQR) ^1^	Post-aCC (IQR)	*p* Value ^2^	Effect Size (*r*)
PTS Calls	27	5 (2–7)	2 (0–4.5)	0.003 **	0.60
ED Presentations	27	15 (8.5–20.5)	4 (0–15.5)	0.014 *	0.48
Mental Health ED Presentations	27	9 (5.5–16.5)	1 (0–7.5)	0.018 *	0.46
Mental Health Admissions	27	3 (1–5.5)	0 (0–1)	0.0005 ***	0.68
Mental Health Episodes	27	8 (3.5–11)	1 (0–3)	0.0000797 ****	0.77
Incidents and Adverse Events	27	3 (0–13.5)	0 (0–4)	0.011 *	0.52

^1^ IQR: Interquartile Range. ^2^ ****: *p* < 0.0001; ***: *p* < 0.001; **: *p* < 0.01; *: *p* < 0.05.

**Table 2 healthcare-13-02391-t002:** Twelve-month pre/post-intervention median service costs for patients discharged from the aCC program.

Measure	*N*	Pre-aCC (IQR) ^1^	Post-aCC (IQR)	*p* Value ^2^	Effect Size (*r*)
Emerg Medical Total Costs	27	7010 (4355–9929)	1998 (0–6044)	0.003 **	0.55
Emerg Nursing Total Costs	27	6773 (4690–10,097)	2247 (0–6085)	0.003 **	0.55
Pathology Total Costs	27	530 (267–1290)	266 (29–486)	0.028 *	0.42
Pharmacy Total Costs	27	1753 (323–3608)	240 (0–588)	0.005 **	0.54
Other Mh Total Costs	27	12,681 (4882–25,809)	3021 (427–9136)	0.001 ***	0.59
Community Total Costs	27	0 (0–0)	0 (0–0)	0.673	0.08
Total Costs	27	64,921 (30,174–141,967)	19,329 (3335–30,713)	0.000184 ***	0.68

^1^ IQR: Interquartile Range. ^2^ ***: *p* < 0.001; **: *p* < 0.01; *: *p* < 0.05.

## Data Availability

The original contributions presented in this study are included in the article. Further inquiries can be directed to the corresponding author.

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
