# Peer review of "Agile Comprehensive Care: A Pragmatic Trial of a Systemic Intervention for High Utilizers of Emergency Departments"

_healthcare, 2025, doi:10.3390/healthcare13192391_

Round 1
Reviewer 1 Report
Comments and Suggestions for Authors
This study provides important information for practice. Authors have reasonably described the study limitations and the implications for limited generalizeability of the findings. As such the conclusions should be restated as '...intervention...may be an effective approach...' vs '...is an effective approach...'.
See additional notations below:
Methods:
- The 'large amounts of missing outcome data' mentioned is critical as it impacts the robustness of the analysis. Authors should indicate or quantify the level of missing data.
- Provide a sample of a CSP from the aCC. This will enhance and support the reproducibility of the study
- What was the response rate of the aCC evaluation? i.e. '25 survey responses received...'
- What was the distribution of responses from across the service types?
Reviewer 2 Report
Comments and Suggestions for Authors
Based on my review of the paper, I suggest following changes.
Providing information on cost reduction pre and post aCC intervention, may be misleading without mention of the reasons for ER visits. Please elaborate if these data was collected and present the data.
Provide information if the reasons for those visits were related to the same conditions. As an example, patient could have pre intervention visits related to mental health issues and appendicitis that was treated with appendectomy. Post aCC intervention, costs related to visits from mental health issues would certainly be less as there is no added cost from appendicitis. Highlight these confounding factors when interpreting cost reductions from the aCC interventions.
Did this patient subclass have frequent visits related to pregnancy related conditions? Elaborate.
Provide data on compliance with aCC clinic visits and how it affected the outcomes.
Elaborate if results were influenced by seasonality. As an example, patients could have frequent ER visits due to respiratory infections during winter months, which would dip during summer months and could be falsely attributed to aCC clinic interventions.
Reviewer 3 Report
Comments and Suggestions for Authors
I appreciate the opportunity to review the paper “High utilizers of emergency departments: agile Comprehensive Care.” The authors aim to assess the association of aCC service utilization and outcomes for 27 patients in a Melbourne, Australia hospital system. Their research design is a retrospective pre-post research design and a survey of staff. The authors find that aCC care was associated with significant reductions in emergency department (ED) visits, including a reduction in mental health ED visits, and a $45,492 reduction in median post-aCC care costs. This paper addresses an important healthcare challenge, and the results demonstrate the clear potential of aCC. I have some suggestions for the authors that would strengthen and clarify the paper.
Comments:
- I urge the authors to be careful about making causal claims when reporting and discussing their results. Throughout the abstract and main text, the authors use causal language, such as "evaluate the effectiveness of the aCC service" and concluding that aCC "demonstrated that a systems' and evidenced based approach results in improved clinical outcomes and reduced service usage and cost." However, their single-group pre-post design cannot establish causation, only association.
Without a control group, randomization, or quasi-experimental design, the observed changes could result from multiple factors, such as natural regression to the mean, other concurrent interventions, or unobserved changes in personal circumstances or healthcare delivery. The authors briefly acknowledge this limitation in their discussion but continue to present results as evidence of intervention effectiveness throughout the paper. Since this is a retrospective analysis that does not require active recruitment, the authors may want to consider identifying a control group of patients to strengthen the analytical approach by using a difference-in-differences research design. - The sample selection criteria introduce bias that limits the study’s generalizability. By design, the study focuses on the most successful aCC patients. Patients had "in-depth file review" by the aCC team to determine suitability and were limited to those deemed likely to benefit from aCC intervention. Ultimately, only 27 patients with complete 12-month follow-up data were included in the sample.
This process is understandable for a high-resource program like aCC but likely biased results toward favorable outcomes since the intervention was only offered to patients the clinical team believed would respond well. I suggest that the authors further clarify the study inclusion criteria, including a discussion of how many patients were excluded from aCC care and what factors shaped these decisions. The study would also benefit from an additional analysis of patient attrition: How many patients were initially registered into the aCC clinic during the period considered? How many patients who were referred or discharged from the clinic did not meet study inclusion criteria? What is the composition of this group? How many patients could be followed for 6 months but not for 12?
- In the discussion of costs, the authors should clarify if the cost reported is per-patient or total for all patients in the sample. Also, I would urge the authors to consider including the aCC cost in the analysis, or at least report it.
- The authors acknowledge "large amounts of missing outcome data" for clinical measures but provide no explanation for why this occurred or a discussion of potential bias from missing data. This should be addressed more thoroughly.
